# What is a good question? Task-oriented asking with fact-level masking

## ABSTRACT

Asking questions is an important element of real-life collaboration on reasoning tasks like question answering. For example, a legal assistant chatbot may be unable to make accurate recommendations without specific information on the user's circumstances. However, large language models are usually deployed to solve reasoning tasks directly without asking follow-up questions to the user or third parties. We term this problem task-oriented asking (TOA). Zero-shot chat models can perform TOA, but their training is primarily based on next-token prediction rather than whether questions contribute to successful collaboration. To enable the training and evaluation of TOA models, we present a definition and framework for natural language task-oriented asking, the problem of generating questions that result in answers useful for a reasoning task. We also present fact-level masking (FLM), a procedure for converting natural language datasets into self-supervised TOA datasets by omitting particular critical facts. Finally, we generate a TOA dataset from the HotpotQA dataset using FLM and evaluate several zero-shot language models on it. Our experiments show that current zero-shot models struggle to ask questions that retrieve useful information, as compared to human annotators. These results demonstrate an opportunity to use FLM datasets and the TOA framework to train and evaluate better TOA models.

## 1 INTRODUCTION

Asking and answering questions are important elements of collaboration when performing reasoning tasks like question answering (QA). Large language models (LLMs) have shown strong performance on question answering (Chang et al., 2023), but little investigation has been done on their ability to ask useful questions during collaboration. In the context of collaboration, a question is useful if its answer helps with the downstream task, like QA. Many LLM applications would benefit from the ability to ask these kinds of intermediate questions.

Consider the use case of a corporate legal assistant chatbot. Providing proper legal advice is often highly situational, even differing from case to case within a single company. Important details such as an executive's level of risk tolerance may never appear in writing, so they must be requested directly from the user. In many commercial products like tax software, static methods such as forms and decision trees are used to request information (Yu et al., 2020). More recently, chatbots have been used to leverage the convenience of dialogue. However, many chatbots (Wang et al., 2019) only generate questions from a hand-coded fixed list, which does not scale or generalize well. The most recent generations of zero-shot chat LLMs are capable of generating questions by predicting the maximum likelihood response. However, this strategy fails to account for non-contextual factors, like what facts may be known by each agent. For example "Did your tax bill exceed $X?" may be much easier for the user to recall than "How much tax did you pay last year?" Thus it is important to evaluate questions based on their effect on the user's goal, rather than their likelihood. To promote objective comparison of models' ability to ask for missing information, we propose task-oriented asking (TOA) as a benchmark for language model question generation.

Existing TOA datasets are few, limited in scope, and do not evaluate the impact of TOA within the context of a complex natural language reasoning task. To evaluate and fine-tune TOA mod-

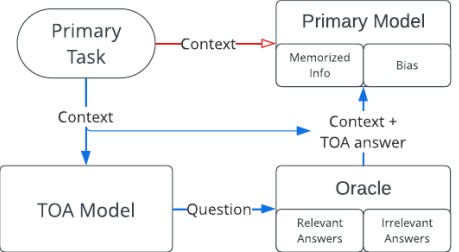

Figure 1: Overview of the TOA task, which simulates the need to formulate a question. Conventionally (⟶▷), the primary model performs the primary task directly. However, in TOA (⟶▶), the TOA model first uses the context to generate an intermediate question. The question is presented to the oracle model, which generates a response. The response is appended to the original context, then finally sent to the primary model. For strong TOA models, we expect the primary model to achieve better performance on context plus TOA answer than on context alone.

els, we propose creating a large, challenging natural language TOA dataset. Each example in a TOA dataset should contain an instance of a primary task, such as QA. Ideally, the example context should be missing some information critical to completing the task, naturally prompting a question seeking that information. Manually creating such information-incomplete tasks is time-consuming, so instead we propose a method for creating these datasets through self-supervision. We begin with an existing QA dataset that contains critical factual information in the example contexts. Then we omit or "mask" portions of the context, naturally creating QA instances lacking sufficient background information. This process has the additional benefit of specifying exactly what information is missing from each example. We term this process fact-level masking (FLM).

These masked contexts provide an opportunity for language models to generate questions that seek out information. To evaluate their success, it would be useful to know what kinds of answers a generated question would elicit. One approach would be to annotate a corpus of question-answer pairs, but this strategy does not scale well, nor generalize to unseen questions. Instead, we propose simulating the answering agent (the "oracle") with a language model that outputs responses dynamically. This allows us to produce responses to a wide range of generated questions without the need for annotation.

Given some response to a question, we would like to evaluate whether it contains useful information originally absent from the context. Again one might consider labeling a corpus of answers as "useful" or "not useful," but this too does not scale well. Whether information is useful depends on many factors, such as whether the information appears elsewhere in the context or whether the consuming agent has previously memorized the information. We propose instead to evaluate the information based on whether the downstream agent (e.g., QA model) consuming the information improves its performance. For example, we can assess the change in QA accuracy of a model with versus without the response. Similar to our process of producing responses, this procedural approach robustly evaluates question-answer pairs. In this way, the primary and oracle models form a pipeline that, together with the FLM dataset, can be used to objectively evaluate the usefulness of an information-seeking question.

In summary, the primary contributions of this paper are threefold:

- A definition and framework for evaluating natural language task-oriented asking (TOA)

- Fact-Level Masking (FLM): A procedure for converting natural language datasets into self-supervised TOA

- FLM-HotpotQA: A TOA dataset generated with FLM and associated evaluation pipeline for FLM-HotpotQA with baseline results [1]

---

[1]We will release our code publicly after acceptance.

We find that although zero-shot LLMs can generate plausible-sounding questions, these questions extract less useful information compared to human-generated questions. This finding indicates that current state-of-the-art LLMs are still limited in their capacity to pose insightful questions, unveiling a need for TOA training and evaluation methods and a promising avenue for future research.

## 2 RELATED WORK

### 2.1 GENERAL QUESTION GENERATION

Question Generation (QG), speaking generally, is the task of automatically generating questions (Rus et al., 2008). Questions can be generated using syntactic (Gates, 2008; Yao et al., 2012) or neural (Chen et al., 2018) approaches. Duan et al. (2017) and Wang et al. (2020) generate questions for data augmentation for QA tasks and pretraining, respectively, using convolutional, recurrent, and transformer architectures. Chatbots designed for social dialogue may ask questions to exhibit emotional intelligence, prompt users, and drive engagement (Shum et al., 2018). Non-task-oriented question-asking can also be used for educational purposes (Kurdi et al., 2020). During general QG, outputs are often evaluated based on the Bleu or Rouge score of the question, as in (Chen et al., 2018), while questions in social dialog are evaluated by the same metrics as other dialog turns (e.g., user engagement (Yu et al., 2016)). Our approach, in contrast, evaluates a question based on the usefulness of the answer it prompts (Figure 1).

### 2.2 TASK-ORIENTED QUESTION GENERATION

TOA is a subset of question generation where a question is evaluated based on whether its answer is useful to a downstream primary task. Clarifying questions are a type of task-oriented question for disambiguating user intent, as in (Aliannejadi et al., 2019). ClariQ (Aliannejadi et al., 2020) is a supervised dataset of queries and clarifying questions for them. The GuessWhat?! (De Vries et al., 2017) and CLEVR Ask (Matsumori et al., 2021) datasets are examples of constrained iterative binary TOA tasks in the vision domain. These datasets contain images paired with conversations by annotators playing a guessing game. In GuessWhat?! one annotator asks questions trying to narrow down which object in the image the other has chosen, and the other answers yes or no. In CLEVR Ask, images and answers are generated synthetically. Existing TOA datasets are sparse and limited in scope to clarifying or binary questions and to the vision domain while we introduce the first natural language TOA dataset in the QA domain.

### 2.3 RELATED TASKS

In task-oriented dialog (TOD), the system is designed to converse with the user to perform a slot-filling task. Slot-filling tasks are typically straightforward and well-defined, like booking a hotel. What information is missing, such as the desired price range, is usually easily defined by which slots are empty (Budzianowski et al., 2018). In such cases, template-based systems can be sufficient for question asking in TOD, with the main challenge being natural language understanding and dialog state tracking. Since the set of useful questions is neither complex nor numerous, TOD systems often assume that the user will be able to answer all system-generated questions. By decoupling TOD from a fixed slot ontology and accounting for incomplete user knowledge, TOA can be viewed as a generalization of the dialog planning and natural language generation steps of TOD.

Finally, TOA is similar to the idea of tool-use, where models can consult APIs like a calculator, search engine, or QA model to improve performance on a downstream task. Tool-use models like Toolformer (Schick et al., 2023) call APIs internally during generation to gather additional knowledge. Tool-use using a QA model differs from task-oriented asking in that TOA allows for independently training and inference of the question generation model.

## 3 METHODS

### 3.1 PROBLEM DESCRIPTION

The goal of task-oriented asking is for the TOA model to transfer information from a knowledge-able agent (the oracle) to an executive agent (the primary model) by asking a question. The primary model is a model that directly executes some task for the user, such as a legal assistant chatbot or QA model. The oracle is an agent capable of answering intermediate questions related to the primary task. This could be a human user, expert, or LLM stand-in like Flan-T5 (Chung et al., 2022). The TOA model is a computer agent, capable of generating questions that assist the primary model in its task, such as Alpaca (Taori et al., 2023). It takes the primary task as input and generates a question for the oracle. The oracle response is concatenated to the original context and then passed to the primary model, giving the primary model access to the information requested in the question. The TOA model's performance is evaluated using the difference between the primary model's performance with and without the oracle's answer.

Many factors affect the extent of TOA performance gains elicited by oracle responses, including the context, the bias of the models, the possible oracle responses, and what information has been memorized by the primary model. Hence, TOA performance can only be assessed in the context of a particular pipeline. Our pipeline, as described above, consists of a primary model, $M_1$, tasked with performing some task, and an oracle, $\Theta$, which responds to questions generated by the secondary TOA model, $M_2$. In the next section, we present a specific $M_2 \rightarrow \Theta \rightarrow M_1$ pipeline and dataset on which to evaluate it.

### 3.2 PROBLEM DEFINITION

Let $t$ be a natural language statement of a task. Let $f_1, ..., f_n$ be natural language facts consisting the context for the task. Let example $x = \{t, f_1, ..., f_n\}$. Let $M_1(x) \rightarrow y$ be a primary model that takes $x$ as input and outputs $y$. Let $M_2(x) \rightarrow q$ be a TOA model that takes $x$ as input and generates a natural language question $q$. Let $R(M_1, x, y) \rightarrow r$ be some reward on which $M_1$ is evaluated, where more positive values are better, such as F-score, accuracy, or negative loss. For brevity, we often omit $M_1$ and $y$.

We say a fact $f$ is supporting if it is believed that $R(x \cup \{f\}) > R(x \backslash f)$. Otherwise we say $f$ is distracting (Yang et al., 2018). Let $\Theta(q) \rightarrow f_r$ be an oracle model that takes $q$ as input and returns a response $f_r$. The TOA task is to create a model $M_2$ that maximizes

$$\Delta r = R(x \cup \{f_r\}) - R(x)$$
$$= R(x \cup \{\Theta(M_2(x))\}) - R(x)$$

One may construct more complex versions of TOA involving multiple missing facts, iterative asking, multiple oracles, or cost functions for different types of questions. In this paper, we limit TOA to the costless, single-mask, single-turn, single-oracle case. Similarly, we do not address the separate task of determining whether a task lacks sufficient context.

## 4 EXPERIMENTS

### 4.1 DATASET

We contribute FLM-HotpotQA, a version of the QA dataset HotpotQA for evaluating task-oriented asking (Yang et al., 2018). HotpotQA is a multi-hop QA reasoning task where each example contains both supporting and distractor facts from Wikipedia as determined by human annotators. We choose reward function $R$ to be the F1 score of the word overlap between the predicted answer and the ground truth answer following the original HotpotQA. Thus $r \in [0, 1]$ and $\Delta r \in [-1, 1]$.

To simplify the primary task, we create a complete example $x^c$ that contains the task and every supporting fact (Figure 2). Next, we apply fact-level masking to each example: From each complete example, we create an incomplete example $x^i$ by randomly selecting one supporting

| t | When was the composer of "Persian Surgery Dervishes" born? |
|---|---|
| $f_1^{sup}$ | Persian Surgery Dervishes is a recording of two live solo electric organ concerts, the first held in Los Angeles on 18 April 1971 and the second in Paris on 24 May 1972, by avant-garde minimalist composer Terry Riley. |
| $f_2^{sup}$ ($f^*$) | Terrence Mitchell "Terry" Riley (born June 24, 1935) is an American composer and performing musician associated with the minimalist school of Western classical music. |
| $f_1^{dis}$ | Thomas Christian David (December 22, 1925 - January 19, 2006) was an Austrian composer, conductor, choral conductor, and flutist. |
| $f_2^{dis}$ | Abdolreza Razmjoo is a composer, arranger and singer Tenor of Iran Kurdish ancestry from Kermansha. |

*Incomplete Example $x^i$*

*Complete Example $x^c$*

*Candidate Oracle Responses*

Figure 2: An example containing a primary task t, supporting facts $f_{1,...,n}^{sup}$, and distractor facts $f_{1...n}^{dis}$. (Additional facts not shown.) We create an incomplete example $x^i$ by masking one supporting fact, $f^*$, chosen at random, from the facts in the complete example $x^c$. Prompted with $x^i$, the TOA model poses a question to the oracle which returns one oracle response $f_r$ from the supporting or distractor facts.

fact, $f^* \in x^c$, to be the masked fact and deleting it from the context: $x^i = x^c \setminus f^*$. The masked fact, along with the distractor facts and the other supporting facts, make up the set of possible responses, $f_r$, the oracle may give. We create the response example $x^r$ by appending $f_r$ to $x^i$, i.e. $x^r = x^i \cup \{f_r\}$. When missing one supporting fact, the primary task becomes substantially more difficult, even for strong zero-shot models like GPT-4 OpenAI (2023) (Figure 6). To benchmark human performance, one author of this paper annotated a Test subset of 400 task-oriented questions.

In general, we expect the complete example $x^c$, which contains every supporting fact, to have the highest possible reward. Meanwhile, we say an example $x$ is improvable if there exists at least one possible response $f_r$ such that $\Delta r(f_r) > 0$. By masking facts in $x^c$ we can decrease the reward on the example, producing an improvable self-supervised example. Note that not all incomplete examples will be improvable, for example, if:

- Two facts contain redundant information
- $M_1$ has memorized knowledge of information in $f^*$
- $f^*$ is mislabeled as supporting
- $x^i$ still allows $M_1$ to make a spurious correlation without $f^*$

It is also possible for $x^i$ to be improved by a response $f_r$ even if $f_r \neq f^*$, if $f_r$ and $f^*$ contain similar information. 27.6% and 28.5% of examples in our Full and Test datasets, respectively, are improvable. We preserve unimprovable examples in the dataset to avoid bias; the primary model may sometimes achieve the correct response through a spurious correlation on $x^i$, but fail to make the spurious correlation on $x^r$. Similarly, the primary model may fail on $x^i \cup \{f^*\}$ but succeed on $x^i \cup \{f_r\}$ if $f_r$ contains more helpful information than $f^*$.

## 4.2 IMPLEMENTATION DETAILS

To generate and evaluate answers to TOA questions, we construct the following pipeline (Figure 3). The TOA model $M_2$ takes an incomplete example $x^i$ as input to generate a task-oriented question $q$. As baselines for $M_2$ we choose GPT-4 (OpenAI, 2023), ChatGPT (OpenAI, 2022), and Alpaca (Taori et al., 2023). We select these models for their strong performance on zero-shot and few-shot tasks. We choose a prompt template for each model by evaluating three zero-shot and three 5-shot in-context prompts on 400 examples from the training dataset (Appendix A.1). We also include a dummy Repeater model among the baselines, which simply returns its input.

Questions generated by $M_2$ are passed to the oracle $\Theta$, also a Flan-T5-Base model, which we choose for its accessibility and strong zero-shot performance on other QA tasks. The oracle model serves as a stand-in for a human expert answering task-oriented questions generated by

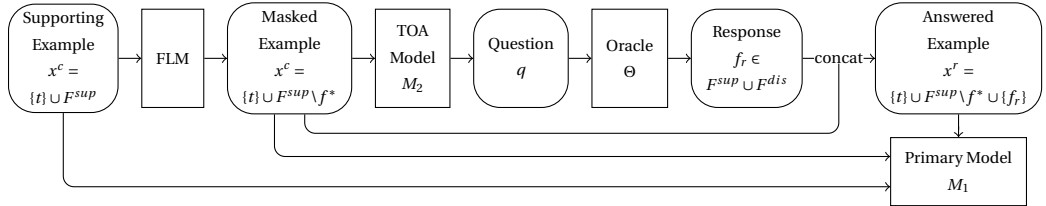

Figure 3: Our TOA evaluation pipeline. From left to right, a complete example is masked via fact-level masking to create an incomplete example. The TOA model generates a question based on the incomplete example. We provide the question to the oracle, which generates a response. The response is concatenated onto the incomplete example to produce the response example. Finally, all three examples are evaluated py the primary model.

Table 1: F1 and exact match recovery for full validation set and manually annotated test subset.

| Model | Full | | | | Test | | | |
|---|---|---|---|---|---|---|---|---|
| | F1 | F1 Recovery | EM | EM Recovery | F1 | F1 Recovery | EM | EM Recovery |
| Alpaca | 61.3 | 39.8 | 45.3 | 38.5 | 60.4 | 39.7 | 46.8 | 45.6 |
| GPT-3.5 Turbo | 59.7 | 30.4 | 44.4 | 31.9 | 59.3 | 34.0 | 45.5 | 38.2 |
| GPT-4 | 65.6 | 64.6 | 49.4 | 66.9 | 65.3 | 65.6 | 50.2 | 66.2 |
| Repeater | 58.3 | 22.5 | 43.1 | 22.8 | 58.5 | 29.6 | 45.8 | 39.7 |
| Human | - | - | - | - | 68.8 | 84.4 | 54.3 | 89.7 |
| Masked | 54.5 | 0.0 | 39.9 | 0.0 | 53.0 | 0.0 | 39.0 | 0.0 |
| Supporting | 71.7 | 100.0 | 54.1 | 100.0 | 71.7 | 100.0 | 56.0 | 100.0 |

$M_2$. $\Theta$ returns $f_r$, the most likely response to $q$ from among all possible distractor facts $F^{dis}$ present in the original HotpotQA example ($\mu = 39.2$, $\sigma = 11.4$), all supporting $F^{sup}$ facts ($n-1$ of which are already present in the context, $\mu = 1.43, \sigma = 0.71$), and the masked fact $f^*$.

To create the response example $x^r$, we append the oracle response to the incomplete example. Note that by appending rather than inserting, the order of facts may be altered as compared to $x^c$, even if $f_r = f^*$, which may occasionally affect the output of the primary model.

Finally, complete examples containing $f^*$, response examples containing $f_r$, and incomplete examples lacking any response are passed to the primary model. If $M_2$ produces a question with positive $M_1$ improvement, then one should expect $R(x^c) \geq R(x^r) > R(x^i)$. To express reward relative to its theoretical minimum ($R(x^i)$) and maximum ($R(x^c)$) values, we define recovery as:

$$\rho = 100 \cdot \frac{R(x^r) - R(x^i)}{R(x^c) - R(x^i)}$$

and select F1 recovery as our primary evaluation metric.

## 5 RESULTS AND DISCUSSION

### 5.1 BASELINE PERFORMANCE

We report F1 and exact match recovery results for the baseline TOA models on the full HotpotQA validation set ($n = 7404$, Figure 4). Of the four baseline models, GPT-4 performs best in both F1 and exact match (EM), recovering 65.6% and 66.9% respectively. These results, however, fall well short of complete recovery of missing information, indicating room for improvement even in strong zero-shot models. Other models perform substantially worse. Alpaca achieved 39.8% F1 and 38.5% EM recovery. GPT-3.5 Turbo (OpenAI, 2022) achieved only 30.4% F1 recovery, which is only a moderate improvement over the dummy Repeater model. We suspect Repeater achieves its recovery (22.5%) by exploiting a bias in the oracle towards choosing responses with high keyword overlap with the input question.

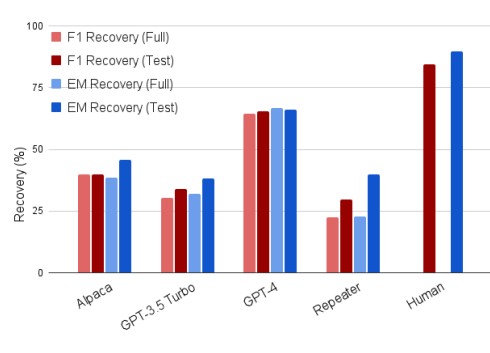

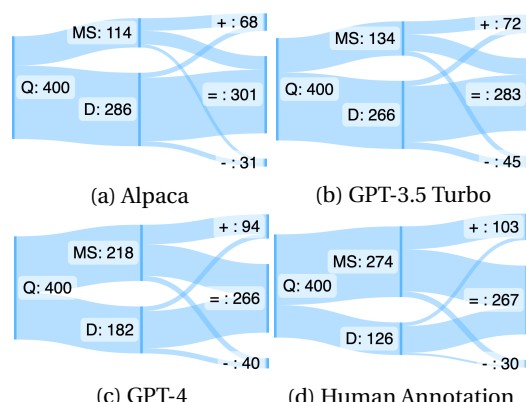

(a) Alpaca     (b) GPT-3.5 Turbo

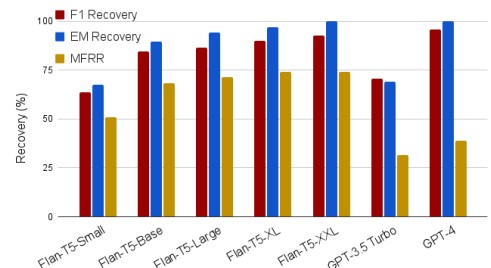

(c) GPT-4     (d) Human Annotation

Figure 4: F1 and exact match recovery of baseline models and human annotators. Results shown for the Full validation set ($n = 7404$) and the Test subset ($n = 400$), which contains human-generated TOA questions.

Figure 5: Proportion of questions (Q) answered with a masked fact (MS) vs. distractor (D) by oracle (left section). Proportion of answers given resulting in positive, zero, or negative difference in primary model performance (right section).

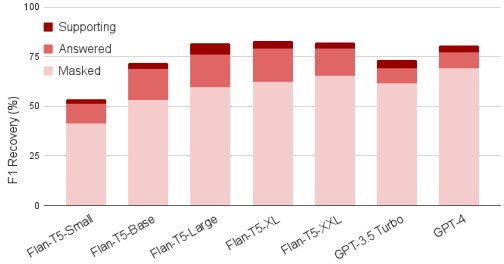

Figure 6: Supporting, answered, and masked F1 as a function of primary model architecture.

Figure 7: F1, exact match and masked fact response rate (MFRR) as a function of oracle size and architecture.

## 5.2 COMPARISON TO HUMAN PERFORMANCE

We find that human-generated questions on the Test subset are more likely to elicit the masked fact $f^*$ in the response. This leads to human annotations performing significantly better than the best baseline models (Figure 5). Human annotation achieved 84.4% F1 and 89.7% EM recovery, compared to the strongest baseline, GPT-4, which achieved 65.6% F1 and 50.2% EM recovery on the test set. This indicates room for improvement in baseline models performing task-oriented asking. Finally, distractor fact responses to human-generated questions were less likely than those by baselines to induce hallucinations in the primary model (6.3% vs. 11.0% of distractors for human generation and GPT-4, respectively).

## 5.3 PRIMARY MODEL ABLATION

We evaluate all available sizes of Flan-T5, GPT-3.5 Turbo, and GPT-4 as candidate primary models using a Flan-T5-Base model as the oracle and human-generated questions as the TOA model. Models lose between 11.7% (GPT-4) and 22.0% (Flan-T5-Large) absolute points F1 score as a result of masking a single supporting fact (Figure 6). Models recover between 65.6% (GPT-3.5 Turbo) and 84.4% (Flan-T5-Base) of the F1 score lost during masking. Although models are affected differently by FLM, with GPT-X models being more robust, consistency in F1 recovery rate suggests that the choice of primary model has minimal impact on TOA evaluation. We suspect GPT-X models are more robust than Flan-T5 since in exploration they appear to have memorized large portions of Wikipedia, which minimizes the impact of removing Wikipedia facts from context.

Table 2: Illustrative example behavior of TOA models on FLM-HotpotQA. Select models omitted for brevity. **Emphasis** added.

(a) Success mode where $f^* \neq f_r$. Despite receiving a distractor response, GPT-4 recovers the missing information contained in the masked fact. Alpaca fails due to requesting irrelevant information.

| Task | Which film was released first: Sacred Planet or Oz the Great and Powerful? | | |
|------|------|------|------|
| Context | Oz the Great and Powerful: Oz the Great and Powerful is a **2013** American fantasy adventure film directed by Sam Raimi and produced by Joe Roth, from a screenplay written by David Lindsay-Abaire and Mitchell Kapner. | | |
| Masked Fact | Sacred Planet: Sacred Planet is a **2004** documentary directed by Jon Long and Hairul Salleh Askor. | | |
| Answer | Sacred Planet | | |

| | Question | Response | $M_1$ Output | |
|------|----------|----------|--------------|---|
| Alpaca | When was Oz the Great and Powerful released? | Snegithiye: The film, released in 2000, proved to be an average grosser at the box office but bagged positive reviews from critics. | Oz the Great and Powerful | |
| GPT-4 | When was the film Sacred Planet released? | Sacred Planet: The film was released by Walt Disney Pictures on April 22, **2004**, and grossed $1,108,356. | Sacred Planet | ✓ |

(b) Failure mode where GPT-4 requests information that is both irrelevant and already contained in the context. The oracle responds to the question accurately, but the new information is not useful, and the primary model predicts "Knoxville, Tennessee," which is incorrect.

| Task | The Tennessee Volunteers football team plays as a member for a conference in what city? | | |
|------|------|------|------|
| Context | 1984 Tennessee Volunteers football team: Playing as a member of the Southeastern Conference (SEC), the team was led by head coach Johnny Majors, in his eighth year, and played their home games at Neyland Stadium in **Knoxville, Tennessee**. | | |
| Masked Fact | Southeastern Conference: The conference is headquartered in **Birmingham, Alabama**. | | |
| Answer | Birmingham, Alabama | | |

| | Question | Response | $M_1$ Output | |
|------|----------|----------|--------------|---|
| GPT-4 | In what city is Neyland Stadium located? | 1983 Tennessee Volunteers football team: Playing as a member of the Southeastern Conference (SEC), the team was led by head coach Johnny Majors, in his seventh year, and played their home games at Neyland Stadium in **Knoxville, Tennessee**. | Knoxville, Tennessee | |
| Human | What city is the Southeastern Conference headquartered in? | Southeastern Conference: The conference is headquartered in **Birmingham, Alabama**. | Birmingham, Alabama | ✓ |

(c) Failure mode where GPT-4 only requests part of the missing information. Both the name and the birth year of the author are missing from the context, which the human annotator successfully requests in a single question. Although GPT-4's question is well-formed, it refers to the manga in question, 'I"s,' overly verbosely as "the manga series featuring 16-year-old high school student Ichitaka Seto," likely causing the oracle to respond with an irrelevant fact. Lacking any years in the context, the primary model hallucinates 1939.

| Task | A Japanese manga series based on a 16-year-old high school student Ichitaka Seto, is written and illustrated by someone born in what year? | | |
|------|------|------|------|
| Context | I"s: The story's main character is 16-year-old high school student Ichitaka Seto who is in love with his classmate Iori Yoshizuki, but too shy to tell her. | | |
| Masked Fact | Masakazu Katsura: Masakazu Katsura (桂 正和, Katsura Masakazu, born December 10, **1962**) is a Japanese manga artist, known for several works of manga, including "Wing-man", "Shadow Lady", "DNA$^2$", "Video Girl Ai", "I"s", and "Zetman". | | |
| Answer | 1962 | | |

| | Question | Response | $M_1$ Output | |
|------|----------|----------|--------------|---|
| GPT-4 | Who is the author and illustrator of the Japanese manga series featuring 16-year-old high school student Ichitaka Seto? | Silver Spoon (manga): Silver Spoon (Japanese: 銀の匙, Hepburn: Gin no Saji) is a Japanese manga series written and illustrated by Hiromu Arakawa, set in the fictional Ooezo Agricultural High School in Hokkaido. | 1939 | |
| Human | When was the author of I"s born? | Masakazu Katsura: Masakazu Katsura (桂 正和, Katsura Masakazu, born December 10, **1962**) is a Japanese manga artist, known for several works of manga, including "Wing-man", "Shadow Lady", "DNA$^2$", "Video Girl Ai", "I"s", and "Zetman". | 1962 | ✓ |

### 5.4 ORACLE ABLATION

We test GPT-3.5 Turbo, GPT-4, and all sizes of Flan-T5 as the Oracle on human-generated questions. Flan-T5-Base and larger respond with the incomplete example in more than 68% of cases (Figure 7). Furthermore, we observe consistently strong performance by these models on F1 and exact match, with both metrics exceeding 84% recovery in all cases. This indicates that when prompted by well-formed and informative questions, Flan-T5 of size Base and larger can consistently respond with appropriate answers. For the sake of accessibility, we choose the smallest strong model, Flan-T5-Base, as our oracle. Interestingly, although GPT-4 responds with the masked fact far less frequently than any Flan-T5 model (GPT-4: 39.0%, Flan-T5-XXL: 74.0%), GPT-4 achieves the highest F1 recovery overall and 100% exact match recovery. This suggests that although GPT-4 gives distractor or redundant supporting facts most of the time, the facts it chooses still carry critical information. This illustrates the importance of measuring information gain rather than nominal correctness.

### 5.5 FAILURE MODES

We observe one failure mode associated with the oracle and three associated with the TOA model, which prevent TOA questions from recovering missing information. Most obviously, the oracle may return an irrelevant and unhelpful response. In 31.5% of cases, human-generated questions induce responses other than the masked fact. When $f^* \neq f_r$, the F1 score of the primary model increases in only 11.1% of cases, compared to 32.5% of cases when $f^* = f_r$ (Figure 5). When a distractor fact does cause an increase in F1, it is often because the distractor fact contains information similar to that in the masked fact (see Table 2a).

Other times, the failure mode is due to the TOA model generating poor questions. For example, in Table 2b, GPT-4 asks a question requesting irrelevant information. The oracle produces an accurate response to the question, but since the information is not useful, the primary model remains incorrect. In other cases, such as in Table 2c, GPT-4 only asks for part of the necessary missing information, requesting the name of the author but not his birth year.

## 6 CONCLUSION, LIMITATIONS AND FUTURE WORK

In this paper, we presented a novel framework for evaluating task-oriented questions and observed that state-of-the-art zero-shot LLMs struggle at this task compared to humans. We attribute this deficiency to a lack of training data and evaluation processes for TOA. To overcome these challenges, we introduced fact-level masking and FLM-HotpotQA, a self-supervised TOA dataset, and an associated evaluation pipeline. To conclude, our contributions highlight the challenges faced by state-of-the-art, zero-shot LLMs, address limitations in training data and evaluation methods for task-oriented asking, and pave the way for future TOA development.

Exploring more challenging task-oriented asking present in normal human interactions is a promising future research direction. Broadly speaking, scope can increase along two axes: the task and the pipeline. One could apply FLM to other tasks requiring longer contexts, more specific domain knowledge, or more hops in logical reasoning. Likewise, increasingly complex tasks such as code generation or chatbot assistants could benefit from TOA. Meanwhile, one may employ an unconstrained generative oracle instead, or employ multiple oracles, each lacking complete information.

Several other possible variations of TOA also have interesting opportunities. Although our work primarily manipulated contextual information, later we will investigate the role of information memorized by the primary agent. This work's modality was also restricted to natural language, but future work may involve tabular, graph, or multi-modal data. In addition, future directions may require iterative task-oriented asking with humans or costly APIs while optimizing each ask. Finally, it remains to be seen how well language models can perform on TOA when specifically trained for the task using self-supervised FLM datasets or reinforcement learning.

### ACKNOWLEDGMENTS

Acknowledgments hidden during the submission phase.

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

## A APPENDIX

### A.1 LIST OF PROMPTS

1. Ask another question that would help you answer the following question: {context} {q1}
2. Some information is missing from this context. Ask a simpler question that would help you answer it. Context: {context} Main Question: {q1} Simpler question:
3. What question can you ask to help you answer the final question? {context} {q1} You can ask:
4. Ask another question that would help you answer the following question: {in-context examples} {context} {q1}
5. Some information is missing from this context. Ask a simpler question that would help you answer it. {in-context examples} Context: {context} Main Question: {q1} Simpler question:
6. What question can you ask to help you answer the final question? {in-context examples} {context} {q1} You can ask:

Based on performance on $n = 400$ examples from the HotpotQA train dataset we select prompts 3, 3, and 2 for Alpaca, GPT-3.5 Turbo, and GPT-4, respectively.

### A.2 ORACLE ARCHITECTURE ABLATION

Table 3: Oracle architecture ablation for oracle models using Flan-T5-Base as primary model on the full validation set.

|  | F1 | F1 Recovery | EM | EM Recovery | MFRR |
|---|---|---|---|---|---|
| Flan-T5-Small | 64.9 | 63.8 | 50.5 | 67.6 | 50.7 |
| Flan-T5-Base | 68.8 | 84.4 | 54.3 | 89.7 | 68.5 |
| Flan-T5-Large | 69.2 | 86.5 | 55.0 | 94.1 | 71.3 |
| Flan-T5-XL | 69.8 | 90.1 | 55.5 | 97.1 | 74.3 |
| Flan-T5-XXL | 70.4 | 92.9 | 56.0 | 100.0 | 74.0 |
| GPT-3.5 Turbo | 66.2 | 70.8 | 50.7 | 69.1 | 31.5 |
| GPT-4 | 70.9 | 95.9 | 56.0 | 100.0 | 39.0 |
| Masked | 53.0 | 0.0 | 39.0 | 0.0 | - |
| Supporting | 71.7 | 100.0 | 56.0 | 100.0 | - |

## A.3 PRIMARY MODEL ABLATION

Table 4: Primary model architecture ablation using Flan-T5 base as oracle on the Full validation set.

| | F1 | | | | EM | | | |
|---|---|---|---|---|---|---|---|---|
| | Masked | Answered | Supporting | Recovery | Masked | Answered | Supporting | Recovery |
| Flan-T5-Small | 41.4 | 51.1 | 53.6 | 79.3 | 28.5 | 35.3 | 37.8 | 73.0 |
| Flan-T5-Base | 53.0 | 68.8 | 71.7 | 84.4 | 39.0 | 54.3 | 56.0 | 89.7 |
| Flan-T5-Large | 59.8 | 76.1 | 81.8 | 74.2 | 42.5 | 58.0 | 63.5 | 73.8 |
| Flan-T5-XL | 62.3 | 78.9 | 82.9 | 80.5 | 45.8 | 60.8 | 64.8 | 78.9 |
| Flan-T5-XXL | 65.2 | 78.9 | 82.2 | 80.6 | 50.5 | 62.5 | 65.8 | 78.7 |
| GPT-3.5 Turbo | 61.4 | 69.3 | 73.5 | 65.6 | 32.5 | 36.8 | 41.3 | 48.6 |
| GPT-4 | 69.0 | 77.2 | 80.7 | 70.2 | 43.0 | 47.3 | 51.2 | 51.5 |

## A.4 RESPONSE FLOW

Table 5: Flow of responses and outcomes for baseline models on the human-annotated Test subset (%). Masked and distractor response indicate how frequently questions elicit masked and distractor facts from the oracle, respectively. +, =, and - indicate whether the masked or distractor fact increased, had no effect, or decreased primary model performance. Distractor hallucination rate indicates how often distractor responses cause a decrease in performance, as a percentage of all distractor responses.

| | Alpaca | GPT-3.5 Turbo | GPT-4 | Repeater | Human |
|---|---|---|---|---|---|
| Masked Response | 28.5 | 33.5 | 54.5 | 32.8 | 68.5 |
| Distractor Response | 71.5 | 66.5 | 45.5 | 67.3 | 31.5 |
| Masked + | 12.3 | 12.8 | 19.0 | 15.0 | 22.3 |
| Masked = | 13.3 | 16.3 | 30.5 | 14.0 | 40.8 |
| Masked - | 3.0 | 4.5 | 5.0 | 3.8 | 5.5 |
| Distractor + | 4.8 | 5.3 | 4.5 | 5.3 | 3.5 |
| Distractor = | 62.0 | 54.5 | 36.0 | 51.7 | 26.0 |
| Distractor - | 4.8 | 6.8 | 5.0 | 10.3 | 2.0 |
| Distractor Hallucination Rate | 6.6 | 10.2 | 11.0 | 15.2 | 6.3 |

