# OpenReview forum: "What is a good question? Task-oriented asking with fact-level masking"
_ICLR.cc/2024/Conference — ICLR 2024 Conference Withdrawn Submission_

### Official Review · Reviewer_egWC · 2023-10-28

**Soundness:** 2 fair
**Presentation:** 2 fair
**Contribution:** 1 poor
**Rating:** 3
**Confidence:** 4

**Summary:**

In this paper, the authors introduce a novel framework for evaluating task-oriented questions, highlighting the difficulties state-of-the-art zero-shot LLMs face in this task compared to human performance. The authors attribute these challenges to a lack of training data and evaluation methods specific to Task-Oriented Asking (TOA). To address these issues, they propose fact-level masking and FLM-HotpotQA, a self-supervised TOA dataset, and an associated evaluation pipeline.

**Strengths:**

1. It is interesting to formulate the follow-up question generation task as a problem of task-oriented asking (TOA).
2. For the TOA task, this paper presents a fact-level masking method to generate TOA dataset, which can be used for training and evaluation.
3. Multiple zero-shot large language models are evaluated on the proposed FLM dataset and show that they struggle to ask a proper question for the given task.

**Weaknesses:**

1. A significant point of contention revolves around the definition of the term "task-oriented." Utilizing a question from HotpotQA as the primary task, denoted as $t$, may not be the most reasonable approach. The task itself could encompass a repetitive and abstract procedure, and task-oriented questions primarily serve the purpose of populating missing values.

2. One noteworthy contribution of this paper pertains to the introduction of the TOA dataset. However, the exposition of this dataset remains somewhat unclear. Key details, such as the dataset's fields and the methodology employed for its measurements, are either absent or challenging to comprehend.

3. In section 4.2, the pipeline directly employs established models like GPT4 and other large language models as M2, with Flan-T5-base serving as $\Phi$. Given that all these models are readily available and widely known, it raises the question of what unique contribution this paper brings to the field.

4. Merely establishing a dataset and making what could be construed as a minor modification to an existing one may not be considered particularly groundbreaking for a conference like ICLR. Furthermore, solely assessing this dataset using existing models might fall short of the standards expected for this conference.

**Questions:**

The major concerns are listed in Weaknesses.

1. Why do we need the TOA task? What benefits can it bring to the QA community? And what is the difference between TOA and the recent tool-use method?
2. How to train a new model using the generated dataset?
3. What are the differences between the TOA task and the proactive dialogue task?

---

### Official Review · Reviewer_Hko3 · 2023-10-30

**Soundness:** 2 fair
**Presentation:** 2 fair
**Contribution:** 2 fair
**Rating:** 3
**Confidence:** 3

**Summary:**

This paper introduces the task of task-oriented asking, targeting the scenarios where language models are used for reasoning tasks that lack necessary contexts. The authors introduce a procedure for converting a QA dataset with multiple supporting facts for each question into a TOA dataset: Randomly mask out a single supporting fact, let a TOA model generate a question that could lead to the fact, and test the model's relative performance gain after adding an accordingly retrieved fact as context. A dataset called FLM-HotpotQA is constructed with this process, and the authors tested GPT-3.5, GPT-4, and Alpaca's performance on the proposed task. Extensive ablation studies on the primary QA model and Oracle model are presented. The authors also showed some failure modes for the task.
Generally, this paper presents an interesting task that current state-of-the-art LLMs may potentially fail at. However, I lean towards a rejection of the paper due to the lack of experiment details, and problems with the oracle design. Details are in the following sections.

**Strengths:**

1. The task of TOA is an interesting testbed, especially for LLMs. Although similar to slot-filling for dialogue systems, the proposed TOA task is more flexible and is potentially better suited for LLMs.
2. The introduced FLM-HotpotQA dataset is useful for evaluating future LLM performance in question generation.
3. Extensive ablation studies on the primary QA model and the oracle model provide insights for understanding the task.

**Weaknesses:**

1. Key implementation details are missing in the paper, which harms the understanding of the paper. It is unclear how exactly different Oracle models select facts based on the questions, and what the prompt for the QA model is.
2. Although the idea of TOA is interesting, the implementation of the task in this paper does not fully reflect its motivation, which is "(TOA) models’ ability to ask for missing information". For example, the involvement of the oracle model in this task makes the actual evaluation of the TOA model tricky. According to Figure 7, the tested oracle models have a low MFRR even with human-generated questions. The low performance of the oracle model is a serious bottleneck in the evaluation, which makes the model potentially impossible to achieve a high performance on the task. The lack of Oracle models' implementation details along with the low performance of the Oracle models harm the soundness of this paper.
3. The results in Figure 6 are doubtful. With supporting facts, the task for the primary model is degenerated to the original HotpotQA. Seeing FLAN-T5-Base on par with GPT-3.5-Turbo and FLAN-T5-Large outperforming GPT-4 on HotpotQA is surprising. The lack of primary models' implementation details along with the surprising results in Figure 6 also harm the soundness of this paper.
4. The related work section could be updated. The area of Asking Clarification Questions (ACQ) is active in dialogue systems research. Please refer to [1].
[1] Rahmani, Hossein A., et al. "A Survey on Asking Clarification Questions Datasets in Conversational Systems." arXiv preprint arXiv:2305.15933 (2023).

**Questions:**

Other than the questions about implementation details, the low performance of the Oracle models, and the surprising results in Figure 6, Below are some more questions for the authors:
1. The procedure of Fact-level Masking (FLM) as a procedure is listed as one of the contributions. Other than HotpotQA, is this process applicable to other datasets?
2. How would you compare the TOA task and dialogue system's ACQ task (especially the generative ones mentioned in the ACQ datasets survey)?
3. What do the $\mu$ and $\sigma$ for $F^{dis}$ and $F^{sup}$ mean in page 6?
4. In section 5.4, you mentioned that although GPT-4 as an oracle model only returns the masked fact 39% of the time, this setting achieves 100% exact match recovery. Does this suggest that a powerful oracle model, which can always choose critical information, might make the TOA model's question generation ability independent of the primary model's recovery rate?

Some minor points:
1. On page 1, there should be references after "Existing TOA datasets are...".
2. On page 5 line 3, $f_r$ here refers to "the set of possible responses" instead of a single fact.
3. Typos: In section 4.2, "also a Flan-T-Base model"->"a Flan-T-Base model"; In the caption of Figure 3, "py"->"by"; In the legend of Figure 4, "Full"->"Val".

---

### Official Review · Reviewer_tQwg · 2023-11-06

**Soundness:** 4 excellent
**Presentation:** 3 good
**Contribution:** 3 good
**Rating:** 6
**Confidence:** 4

**Summary:**

This work introduces a framework, task-oriented asking (TOA), for language models to ask follow-up questions to clarify downstream tasks. To evaluate this framework, this work introduces a QA dataset derived from HotPotQA, constructed by masking one of the supporting fact for answering the question, and expecting TOA to generate a question soliciting the supporting fact. TOA is then evaluated on its ability to ask useful clarification questions which improve performance on the task upon receiving an oracle answer.

**Strengths:**

1. This work addresses an (understudied) problem of asking follow-up/clarification questions, and contributes a novel framework for addressing the problem, within the realm of QA.

2. Overall, this paper is also quite well-written, and very clearly lays out its method.

3. The paper very comprehensively articulates its scope and limitations, making note of any caveats where they may arise, and also makes the right comparisons and baselines to try and address any limitations in the evaluation (e.g. comparing against the repeater baseline to address limitations laid out in bullets on page 5). It seems like the authors have worked hard to address any spurious correlation or potential biases that may affect the evaluation. Consequently, the results are quite convincing.

3. The paper also provided very comprehensive ablation studies in section 5, and provided concrete examples of failure cases.

4. The paper explores its TOA method with lots of different models, including open-source models.

**Weaknesses:**

1. The current setup in the paper seems somewhat dataset-specific, and the evaluation is also currently only focused on a single task & dataset. In the introduction, the paper frames TOA as a more generic technique for all tasks (e.g. intro makes reference to legal tasks and states “we propose task-oriented asking (TOA) as a benchmark for language model question generation”, without narrowing the scope to just QA tasks with a single missing supporting fact.) Thus, either the claims in the introduction need to be tempered, or the current evaluation scheme should be broadened to give a sense of how well TOA generalizes to other tasks.

2. More specifically, the answer model is currently restricted to picking between a limited set of facts (one of them being the masked supporting fact necessary to answer the original question, and the others being distractor facts), which likely overestimates performance compared to an answering
    1. While understandably the authors were trying to simulate an oracle answer model, note that this does not necessarily tell us how well the question-asking model is, and does necessarily simulate a “perfect answerer”. In particular, the task for the question-asking model shifts from “ask a question seeking, specifically, the missing information” to “ask a question that privileges the masked supporting fact as an answer over any of the provided distractor facts”. In the latter case, we don’t even need to guarantee that the question being asked is comprehensible, or would organically be answered with the supporting fact, but simply that the supporting fact seems like a marginally better response than any of the distractors. For example, it could be the case that the input itself carries information about what information is missing and the question was unnecessary, or the question isn’t actually asking for the missing information / only asks for part of the missing information but the masked fact is still the most pertinent answer.
    2. While comparing against the Repeater baseline takes care of some of these concerns, this still does not take away from the fact that there are factors that aren’t explored due to the answer setup. For example, how comprehensible & easy to answer are questions? Would they naturally lead to an answer that contains the correct fact, supposing when didn’t have a constrained set of possible answers? Answering these questions are important if we’d want to generalize beyond the setup here, as generally we do not have access to a set of possible answers.
    3. Indeed, one of the key challenging considerations of asking questions is that the model needs to ask not just a question that recovers the missing information — but recovers the minimal unit of information that is necessary to perform the end-task. Otherwise, we can imagine a question like “tell me everything you know” being maximally useful under the recovery metric for all inputs. The current task setup is unable to measure whether the questions

3. Related to the above, the paper claims that TOA is able to “generate plausible-sounding questions”. It would be great to get empirical concrete evidence of this — perhaps through some evaluation of the wellformedness of the resulting questions.

4. There were some places in the description of the evaluation framework that were unclear / missing important details (see questions).

5. Can you report error bars for Figure 4?

**Questions:**

1. What is the distinction in annotation: “full” validation set vs. “manually annotated” test set?

2. Some more clarity in the metric / evaluation section (section 4.2) would be useful:
    1. What do mu and sigma represent in the first paragraph of page 6?
    2. Assuming that “F1 recovery” means R=F1 in the equation on page 6, and “EM recovery” means R=EM, but this was not clearly stated anywhere.
        1. Based on this definition, why isn’t recovery of the Repeater baseline at 0? My understanding is that repeater simply repeated the (incomplete) facts in the input? Is it that the Repeater model still uses the oracle to pick an arbitrary fact while the “masked” and “supporting” simply condition on the masked input and full input respectively? This should all be more clearly articulated and the “masked” and “supporting” baselines clearly defined.

3. How well does TOA compare against a baseline that simply prompts the model to “reason step-by-step”, i.e. having it (implicitly) generate missing facts without asking any intermediate questions?

---

> ### Comment · Reviewer_tQwg · 2023-11-14
> **Accidentally mixed up reviews for papers**
>
> Hello authors -- it appears I mixed up the reviews for my papers. I have updated this review to the one for your paper. Apologies for the confusion